# Thermal Energy Analysis of Projectiles during Ricochetting Using a Thermal Camera

**DOI:** 10.3390/ma15134693

**Published:** 2022-07-04

**Authors:** Marcin Jasinski, Krzysztof Szczurowski, Adam Wisniewski, Przemyslaw Badurowicz, Tadeusz Bartkowiak, Norbert Tusnio

**Affiliations:** 1Warsaw University of Technology, Institute of Vehicles and Construction Machinery Engineering, Narbutta 84, 02-524 Warsaw, Poland; krzysztof.szczurowski@pw.edu.pl; 2Military Institute of Armament Technology, Prymasa Stefana Wyszyńskiego 7, 05-220 Zielonka, Poland; wisniewskia@witu.mil.pl (A.W.); badurowiczp@witu.mil.pl (P.B.); 3Center of Shooting Technology, Shooting Ranges-Design-Construction-Equipment, Naftowa 2H, 65-705 Zielona Gora, Poland; tadeusz@tebbex.pl; 4The Main School of Fire Service, Faculty of Safety Engineering and Civil Protection, Juliusza Slowackiego 52/54, 01-629 Warsaw, Poland; ntusnio@sgsp.edu.pl

**Keywords:** thermal energy, ignition initiation, projectile ricochet, thermal camera

## Abstract

This paper presents the results of a study of the hazards of ground ignition and/or explosion when various small-calibre projectiles struck various solid materials placed on a test stand in environments at risk of ignition (fire) or explosion (ricochets and projectile penetration of obstacles). For projectile ricochetting tests, the following were used: an armour plate, concrete, sidewalk and granite slabs, etc., and various small-calibre projectiles: 7.62 × 51 mm SWISS PAP, 7.62 × 51T, 7.62 × 51 mm M80, 7.62 × 54R B-32, 7.62 × 54R LPS and .308 Win. Norma Ecostrike. Projectiles impacts were recorded with a high-speed camera (50,400 fps) and thermal cameras (660 fps) and (2615 fps). The ignition capability of solid flammable materials during projectile ricochetting was studied, and the temperatures and surface areas of isotherms were measured as a function of time. From the spherical distribution of thermal energy radiation in space, their volumes, masses of air occupying the studied area, masses of projectile disintegrating into fragments (after impact), thermal energies during projectile ricochetting, histograms of area temperatures and temperatures were calculated. This energy was compared with the minimum ignition energy of the selected gases and liquid vapours, and the ignition temperature were determined. The probabilities of some of the selected gases and liquid vapours which can ignite or cause an explosion were determined. The thermal energies of the 7.62 × 54R B-32 (3400–9500 J) and 7.62 × 51T (2000–3700 J) projectiles ricochetting on the Armox 600 plate was sufficient to ignite (explode) propane−butane gas. The thermal energy of 7.62 × 54R B-32 projectiles ricochets on the non-metallic components (800–1200 J) was several times lower than that of projectiles ricochets on an Armox 600 plate (3400–9500 J). This is due to the transfer of much of the kinetic energy to the crushing of these elements.

## 1. Introduction

Ricochetting of small-calibre projectiles during fights (soldiers, police and criminals, including terrorists), shooting on firing ranges and hunting is a common phenomenon. It is particularly dangerous in the case of such shooting in built-up areas, where the projectiles ricochetting on the metal structures, as well as building elements (stone, concrete, granite, etc.), may not only injure people, but also destroy the surrounding infrastructure-buildings and technical installations [1,2,3,4,5,6,7]. In addition, ricochetting projectiles in dry and warm terrain can cause ignition and fire of the area, such as mulch and peat [8,9,10,11,12,13,14,15,16,17,18,19,20,21,22,23]. The most dangerous is ricochetting of projectiles on sensitive and dangerous materials, i.e., tanks and installations (refineries, gas and oil transfer stations, etc.) with flammable gases such as propane−butane and methane or with flammable liquids with gasoline, acetone, solvents, etc. (installations, warehouses and stores) or when ricochetting projectiles hit the above tanks and installations.

The United States Department of Agriculture carried out research on the capability of various flammable substrates to ignite from ricochetting projectiles [8]. For the tests, the following ammunitions were used: 7.62 × 39 mm, 5.56 × 45 mm, 7.62 × 51 mm and 7.62 × 54R, with a steel core, lead core and solid copper and with a steel and copper jacket. The projectiles hit the steel plate and debris fell into the box below. The box contained flammable materials of organic origin (dried moss or peat). The main conclusion of the research was that projectiles containing steel elements (core and jacket) and solid copper projectiles have the greatest capability to initiate a fire and the temperature of their debris can reach 800 °C. The article presents the method and results of research on phenomena during small arms shooting, i.e., ricochetting of different projectiles on different substrates in environments threatened by ignition (fire) or explosion and penetration by projectiles of obstacles making barriers separate materials constituting fire hazard for the environment. The knowledge of the results of such tests is important for combat in the hazardous locations listed above. The paper is part of work described also in [24]. The authors presented the process of developing a numerical FEM model of the 5.56 × 45 mm SS109 projectile used to calculate the temperatures occurring during the impact on the steel plate.

For the tests of projectiles ricochetting, we used a stand enabling us to control the parameters of the solid, liquid, gas or vapour environment before and after ignition and the ballistic parameters of projectiles on the flight path immediately before ignition initiation and to obtain the material for possible criminological tests.

Based on the research, the amounts of thermal energy generated by the ricochetting of projectiles on various materials were estimated in order to determine the probability of ignition or explosion when such a projectile is in a flammable atmosphere. The conducted literature review reveals a deficit of this type of studies. Similar studies were carried out in the work [8], but no thermal energy calculations were performed as in this paper.

## 2. Experiment

### 2.1. Projectile Ricochetting on an Armox 600 Armour Plate

Projectile ricochetting tests were conducted in a closed tunnel. For the tests, the following materials were used: a board mounting stand (Figure 1 and Figure 2), an armour plate 500 × 500 × 10 mm Armox 600 (simulating ricochetting of the projectile from the armoured vehicle), aramid fabric intercepting ricochetting projectiles, Photron Fastcam SA-Z 2100K (Manufacturer: Photron, 21F, Jinbocho Mitsui Bldg. 1-105 Kanda Jimbocho Chiyoda-ku, Tokyo, Japan. Sourced from: BRJ sp. z o.o. Janka Muzykanta 4, Warsaw, Poland) high-speed camera, lighting, FLIR X6580sc and FLIR A6901scSLS thermal cameras, camera covers, ballistic barrels of calibres, i.e., 7. 62 × 51 mm (.308 Win.), 7.62 × 54R and 5.56 × 45 mm (.223 Rem.).

During the tests, the temperature of projectiles ricochetting (upwards) on an armour plate set at an elevation angle of 45° was measured. The projectiles ricochetting was recorded with cameras against a 400 × 300 mm measuring frame with 50 × 50 mm “windows”.

The following projectiles were used for testing: 7.62 × 51 mm SWISS PAP, 7.62 × 51T, 7.62 × 51 mm M80, 7.62 × 54R B-32, 7.62 × 54R LPS,.308 Win. Ecostrike standard, 5.56 × 45 mm HC (SS109) and 5.56 × 45 mm M193. Figure 3, Figure 4 and Figure 5 show examples of projectile ricochetting results obtained from cameras (with different recording speeds depending on the measurements): high-speed Photron Fastcam SA-Z 2100K (50,400 fps; a resolution of 896 × 448 pixels) allowed a precise observation of the phenomenon, thermal X6580sc (220 fps or 660 fps; a resolution of 640 × 512 pixels) and thermal A6901scSLS (1000 fps, 1920 fps or 930 fps; a resolution of 640 × 512 pixels) allowed accurate temperature measurement during projectile impact.

### 2.2. Projectile Ricochetting on Non-Metallic Surfaces

Projectile ricochetting tests on non-metallic components were performed on an open range. A rack on which the projectile ricochetting plates were placed; an anti-ricochet shield preventing the shrapnel from retreating to the shooting stand, a Photron Fastcam SA-Z 2100K high-speed camera (observation of projectile impact), FLIR X6580sc and FLIR A6901scSLS thermal cameras (measuring the temperature of the phenomenon) and a 7.62 × 54R calibre ballistic barrel were used for the tests (Figure 6).

The purpose of this study was to test the capability of a solid natural flammable substance, including forest mulch, to ignite from ricochetting projectiles and to measure their temperature. Non-metallic elements (slabs-concrete sidewalk, granite gray, granite black, stoneware on concrete slab and concrete block) were set at an angle of 30° between normal to the slab and horizontal, causing the projectile to ricochet downward into a steel bowl of 200 × 200 × 30 mm, which contained 30 mm-thick forest mulch, consisting of dried small leaves, grass and conifer needles. A 500 × 400 mm metal measuring frame with 50 × 50 mm “windows” was used as a scale.

The following materials with different thicknesses were used to study the ricochets of 7.62 × 54R B-32 and 7.62 × 54R LPS projectiles: a concrete sidewalk slab (70 mm; without reinforcement), a grey granite slab (60 mm), a concrete block (120 mm; without reinforcement), stoneware (8 mm) on a concrete slab (40 mm; without reinforcement) and a black granite slab (57 mm). Projectile ricochets were recorded with cameras (Figure 7, Figure 8 and Figure 9): high-speed Photron Fastcam SA-Z 2100K (50,400 fps), thermal X6580sc (660 fps) and thermal A6901scSLS (2615 fps or 2350 fps and 1005 fps).

## 3. Experimental Results and Analysis

Based on the selected images from thermal camera videos, the following were determined: the temperature distribution during projectile ricochetting, the surface area of given isotherms and the duration of flash as a result of friction of ricochetting projectile on the Armox 600 armour plate or on the surface of non-metallic elements and the burning of the pyrotechnic mass.

The algorithm for calculating the mass of a “cloud” of heated air and projectile debris from thermal and high-speed camera images is shown in Figure 10.

The next steps of the algorithm in National Instruments’ LabView program are as follows:Retrieving an image from a thermal camera;Converting the image from Red−Green−Blue (RGB) to Hue−Saturation−Lightness (HSL) palette and selecting the brightest colours in the tonal range;Calculating the area occupied by selected colours in the image in pixels;Converting (scaling) pixels to millimetres.

From the analysis of the high-speed and thermal camera videos, two shapes of thermal energy propagation were distinguished: semi-cone (Figure 11a) and semisphere (Figure 11b); hence, the algorithm was divided into two paths: semi-cone and semisphere.

The next steps of the algorithm for semi-cone-shaped thermal energy propagation are as follows:Taking an image from a high-speed camera;Measuring the height of the semi-cone (scaling the pixels to millimetres accordingly);Calculating the radius of the semi-cone based on the area calculated for the thermal camera (triangle);Calculating the volume of the semi-cone (based on the height and radius of the semi-cone);Calculating the mass of the air (based on its volume and density) and adding the mass of the projectile, assuming that the projectile disintegrates into a cloud of hot debris in the shape of a semi-cone of thermal energy.

The next steps of the algorithm for thermal energy propagation in the shape of a hemisphere are as follows:Calculating the radius of the hemisphere based on the area of the circle calculated for the thermal camera;Calculating the volume of the hemisphere based on the radius of the hemisphere;Calculating the mass of the air (based on its volume and density) and adding the mass of the projectile, assuming that the projectile disintegrates into a cloud of hot debris in the shape of a hemisphere of thermal energy.

The thermograms obtained from the thermal camera films were used to determine the temperature rise during ricochetting and the duration of the maximum temperature. Example thermograms are shown in Figure 12 and Figure 13.

In Figure 12, there was one temperature peak caused by projectile impact on the Armox 600 armour plate. In Figure 13, there were two temperature peaks with the first, larger one caused by ricochetting on the plate and the second, smaller one caused by ricochetting on the elements attaching this plate to the stand. The temperature rise caused by the impact of the projectile on the Armox 600 plate in Figure 12 and Figure 13 occurred faster than the temperature drop caused in turn by the heat transfer from the projectile fragments to the atmospheric air, i.e., cooling in air.

The maximum temperatures recorded with the thermal cameras are shown in Table 1. Two thermal cameras FLIR X6580sc and FLIR A6901scSLS were used for temperature measurements. The differences in the maximum temperatures for several shots with the same projectile are due to the different ranges of the measured temperatures set. In most cases, the duration of these maximum temperatures was <20 ms. Due to the short duration of the maximum temperature and the relatively low recording speed, the temperature recorded with one camera can be four times higher than with the other.

The final step is to calculate the thermal energy when the projectile ricochets from the following equation:(1)Q=m·cw·Δt,
where *m* is the mass; *t* is the temperature (°C); *cw* is the specific heat in J/(kg °C) calculated from the mass fractions of air and projectile.

Table 2 presents the results of calculating thermal energy during projectile ricochetting on the Armox 600 armour plate, and Table 3 presents the calculation results of thermal energy during projectile ricochetting on the surface of non-metallic components.

The presented calculations showed that in the case of projectile ricochetting on Armox 600 (Table 2) of 7.62 × 51 mm M80, 7.62 × 51 mm SWISS PAP and 7.62 × 54R LPS projectiles, the maximum thermal energies of ricochetting were 35%, 51% and 64% of the projectile kinetic energy, respectively.

However, in two cases of 7.62 × 54R B-32 (3400–9500 J) and 7.62 × 51T (2000–3700 J) projectiles, the thermal energies due to the burning of the incendiary mass of the projectiles were several times higher than the thermal energies of the projectiles: 7.62 × 54R LPS (1100–2100 J), 7.62 × 51 mm SWISS PAP (1600–2000 J) or 7.62 × 51 mm M80 (740–1120 J).

In the case of the 7.62 × 54R LPS projectile ricochets on the surface of non-metallic elements (Table 3), the thermal energy was similar to that of the projectiles ricochetting on the Armox 600 plate, while for the 7.62 × 54R B-32 projectiles ricochets on the surface of non-metallic elements, its thermal energy (800–1200 J) was several times lower than that of the projectiles ricochetting on the Armox 600 plate (3400–9500 J). This is due to the transfer of much of the kinetic energy to the crushing of non-metallic elements (concrete slabs, granite, etc.).

## 4. Estimated Probability of Ignition When a Projectile Ricochets in a Flammable Atmosphere

The thermal energy calculated in Section 3 can be compared with the minimum ignition energy (at different flashpoints) for the selected flammable gases and liquid vapours (Table 4).

It can be seen from Table 2 and Table 3 that during the ricochetting of projectiles in a flammable atmosphere the thermal energy was much higher (800–1200 J) than the minimum ignition energy (0.018–300 mJ). In contrast, in a large number of measurements, the ignition temperature was lower or at the flammable limit, and the duration of the energy and temperature pulse was very short—less than 20 ms. Therefore, an ignition (fire) or explosion hazard can be expected for substances such as acetylene, automotive gasoline or diesel fuel.

## 5. Conclusions

Based on the results of the ignition of natural flammable materials due to the ricochetting of various small-calibre projectiles on various metallic and non-metallic solid surfaces placed on the test stand, the following conclusions can be drawn:When ricochetting on the Armox 600 plate of projectiles:7.62 × 51 mm M80, 7.62 × 51 mm SWISS PAP and 7.62 × 54R LPS maximum ricochet thermal energies were 35%, 51% and 64% of the kinetic energy of the projectile, respectively;7.62 × 54R B-32 (3400–9500 J) and 7.62 × 51T (2000–3700 J) thermal energies due to the burning of the incendiary mass of the projectile were several times higher than the thermal energy of the projectiles: 7.62 × 54R LPS (1100–2100 J), 7.62 × 51 mm SWISS PAP (1600–2000 J) or 7.62 × 51 mm M80 (740–1120 J);The thermal energies of the 7.62 × 54R B-32 and 7.62 × 51T projectiles ricochetting on the Armox 600 plate were sufficient to ignite (explode) propane−butane gas in an atmosphere of the minimal oxygen to cause the ignition of that gas;The capability of projectiles to ignite (explode) propane−butane gas depended not only on the velocity of the projectiles, but mainly on their design and, in particular, on the types and incendiary masses of the projectiles;When ricochetting 7.62 × 54R B-32 projectiles on the surface of non-metallic components, its thermal energy (800–1200 J) was several times lower than when ricochetting the projectile on the Armox 600 plate (3400–9500 J). This is due to the transfer of much of the kinetic energy of the projectile to the crushing of these elements (concrete slabs, granite slabs, etc.);The thermal energy of 7.62 × 54R LPS projectiles ricochetting on the surface of non-metallic components was similar to the thermal energy of projectiles ricochetting on the Armox 600 armour plate;The algorithm developed in National Instruments’ LabView software for calculating the mass of a cloud of heated air and debris correctly describes the thermal energy of this cloud as a result of the following:Converting the image from RGB to HSL and selecting the brightest colours in the tonal range;Converting (scaling) pixels into millimetres;Calculating the area occupied by selected colours in the image in pixels;The algorithm correctly describes the propagation of thermal energy in the shape of:A semi-cone for 7.62 × 51 mm SWISS PAP projectiles as a result of the following:Measurement of the height of the semi-cone (based on the high-speed camera image after scaling the pixels to millimetres);Calculating the radius and volume of the semi-cone (based on the area of the triangle calculated for the thermal camera);Calculating air mass (based on air volume and density) and adding the mass of hot projectile debris in the shape of a semi-cone of thermal energy;Hemisphere for 7.62 × 51T projectiles as a result of the following:Calculating the radius and volume of the hemisphere (based on the area calculated for the thermal camera);Calculating the air mass (based on air volume and density) and adding the mass of hot fragments of the hemisphere-shaped projectile thermal energy;The thermogram of a ricocheting 7.62 × 51T projectile on an Armox 600 plate showed that the time to sustain a fairly high temperature, above 825 °C, was very short at about 14 ms;The thermogram of the ricocheting of 7.62 × 51 mm SWISS PAP projectiles on the Armox 600 showed that there were two temperature peaks. The first, higher peak was caused by ricocheting on the Armox 600 armour plate, and the second, lower peak was caused by ricocheting on the elements fixing this plate. It is, therefore, necessary to install metal plates or non-metallic elements on the stand, on which the projectiles ricochet, so that the predicted surface of ricocheting projectiles, depending on their angle of incidence, does not include (usually steel) installation elements, such as flat bars bolted to the stand;The thermogram showed that the temperature rise due to the impact of the projectile on the Armox 600 armour plate occurred faster than the temperature fall due to the heat transfer from the projectile fragments to ambient air, i.e., cooling in air.The differences in the obtained maximum ricochet temperatures for several shots with the same projectile are due to the set different ranges of measured temperatures, and in most cases, the duration of these maximum temperatures was <20 ms.

## Figures and Tables

**Figure 1 materials-15-04693-f001:**
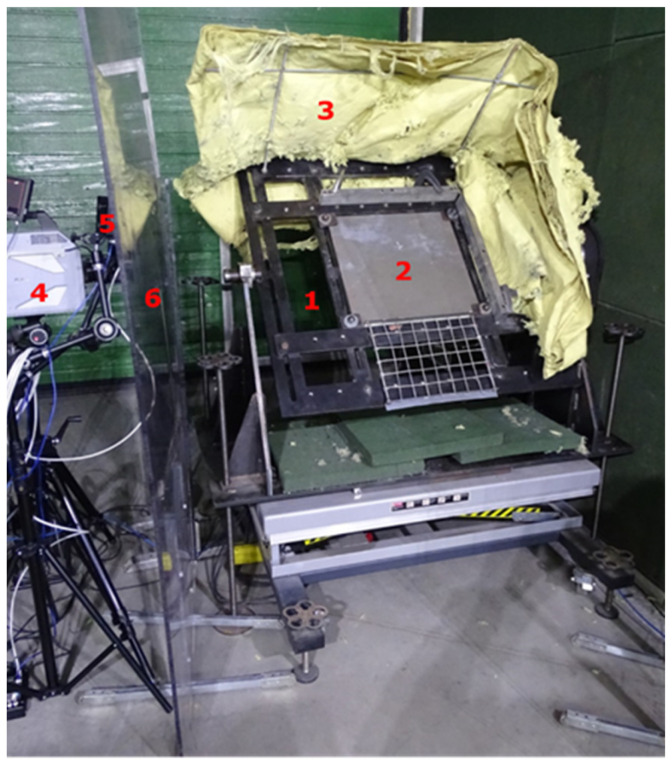
Temperature test stand for ricochetting projectiles. 1, stand; 2, 500 × 500 × 10 mm Armox 600 plate; 3, aramid fabric; 4, Photron Fastcam SA-Z 2100K high-speed camera; 5, lighting; 6, camera and lighting covers.

**Figure 2 materials-15-04693-f002:**
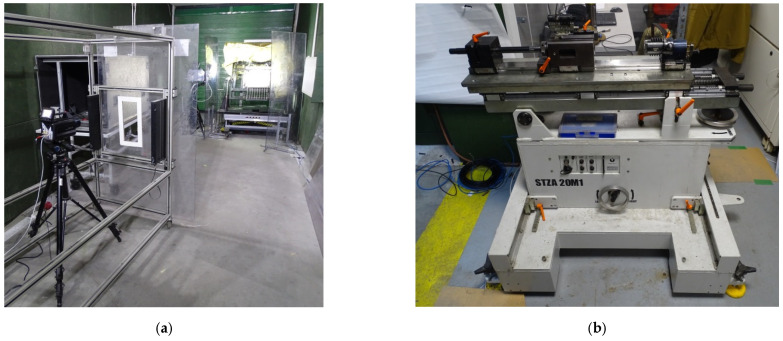
Stand for the temperature testing of ricochetting projectiles: (**a**) FLIR X6580sc thermal camera; (**b**) tripod with a 7.62 × 51 mm ballistic barrel.

**Figure 3 materials-15-04693-f003:**
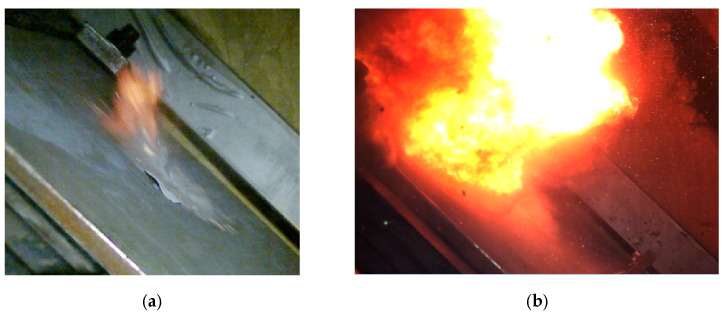
Recording of a fragmented projectile striking an Armox 600 armour plate (Photron Fastcam SA-Z 2100K-50,400 fps): (**a**) 7.62 × 51 mm SWISS PAP projectile; (**b**) 7.62 × 51T projectile flash caused by the rapid combustion of the streaker pyrotechnic mass.

**Figure 4 materials-15-04693-f004:**
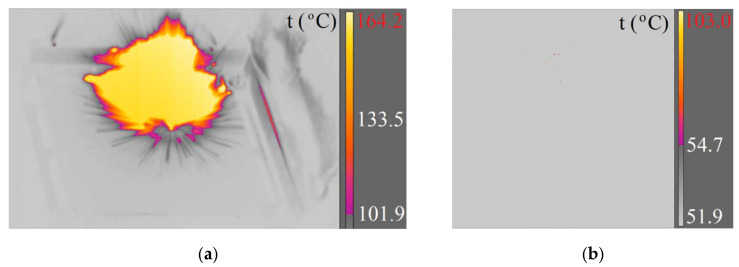
Recording of a 7.62 × 51 mm SWISS PAP projectile fired from a 7.62 × 51 mm ballistic barrel: (**a**) X6580sc camera (220 fps); (**b**) A6901scSLS camera (1000 fps).

**Figure 5 materials-15-04693-f005:**
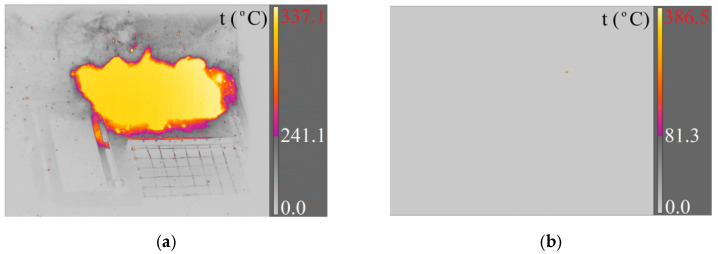
Registration of a 7.62 × 51T projectile fired from a 7.62 × 51 mm ballistic barrel: (**a**) X6580sc camera (220 fps); (**b**) A6901scSLS camera (1000 fps).

**Figure 6 materials-15-04693-f006:**
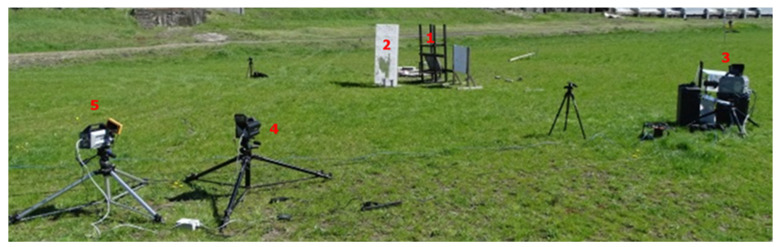
Stand for testing the ignition capability of solid flammable substances together with ricochet temperature measurement. 1, stand with a plate; 2, ricochet shield; 3, Photron Fastcam SA-Z 2100K high-speed camera; 4, FLIR X6580sc thermal camera; 5, FLIR A6901scSLS thermal camera.

**Figure 7 materials-15-04693-f007:**
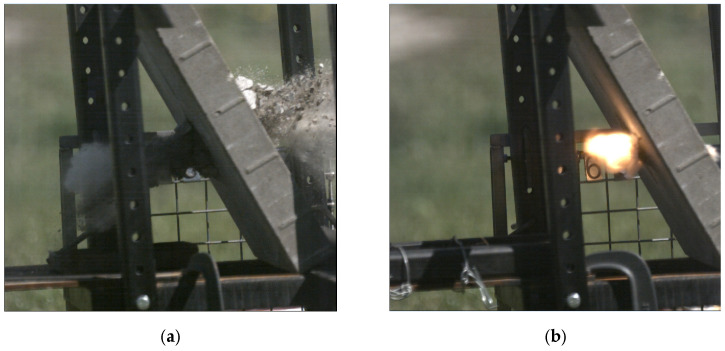
Recording of piercing a concrete sidewalk slab (Photron Fastcam SA-Z 2100K with a speed of 50,400 fps) with the following projectiles: (**a**) 7.62 × 54R LPS; (**b**) 7.62 × 54R B-32 with a visible combustion of the projectile ignition mass.

**Figure 8 materials-15-04693-f008:**
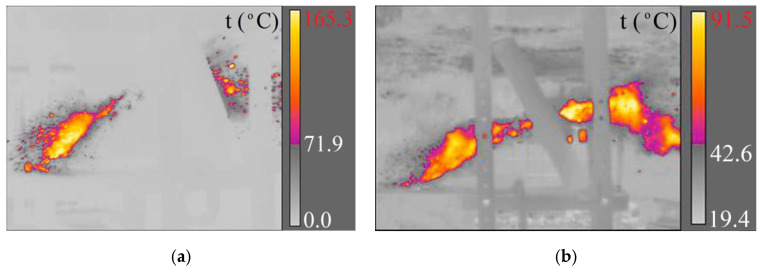
Piercing a concrete sidewalk slab with a 7.62 × 54R LPS projectile recorded with cameras: (**a**) X6580sc (660 fps); (**b**) A6901scSLS (2615 fps).

**Figure 9 materials-15-04693-f009:**
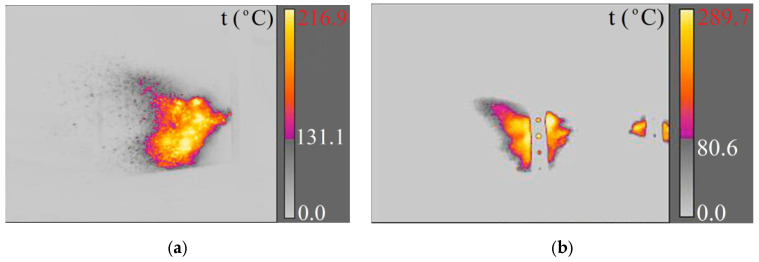
Piercing a concrete sidewalk slab with a 7.62 × 54R B-32 projectile recorded with cameras: (**a**) X6580sc (660 fps); (**b**) A6901scSLS (2615 fps).

**Figure 10 materials-15-04693-f010:**
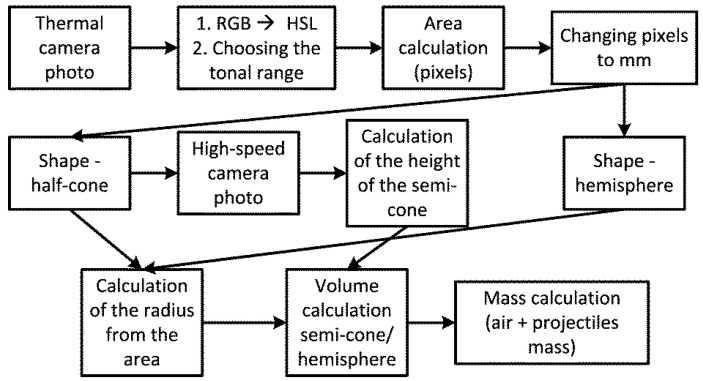
Algorithm for calculating the mass of a cloud of heated air and debris.

**Figure 11 materials-15-04693-f011:**
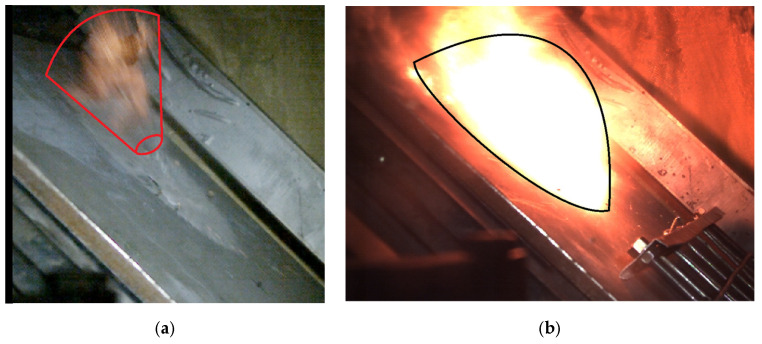
(**a**) Semi-conical shape of the thermal energy propagation of the 7.62 × 51 mm SWISS PAP projectile; (**b**) Hemispherical shape of the thermal energy propagation of the 7.62 × 51T projectile.

**Figure 12 materials-15-04693-f012:**
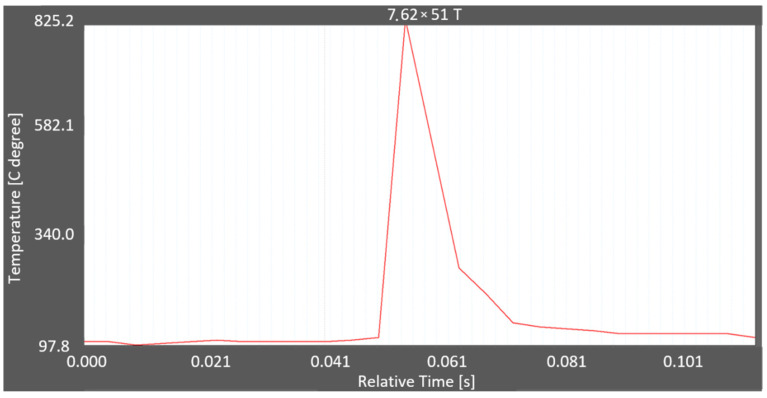
Thermogram of the ricochet of the 7.62 × 51 mm T projectile on the Armox 600 plate.

**Figure 13 materials-15-04693-f013:**
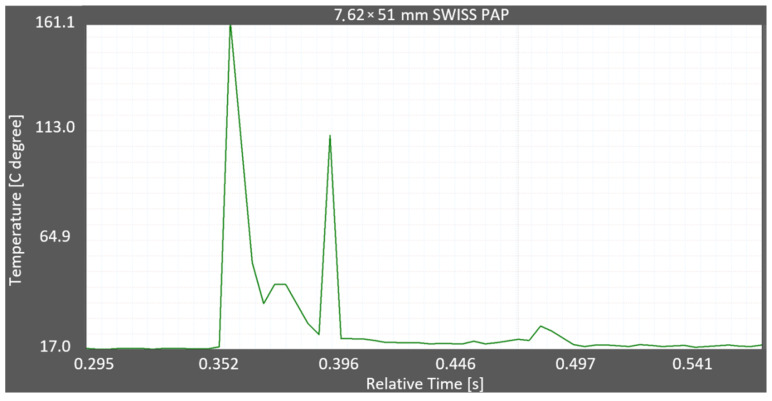
Thermogram of the ricochet of the 7.62 × 51 mm SWISS PAP projectile on the Armox 600 plate.

**Table 1 materials-15-04693-t001:** Maximum projectile ricochet temperatures on the armour plate Armox 600.

Shot Number	Weapon	Ammunition	Maximum Temperature of Gases FLIR X6580sc (°C)	Maximum Temperature of Gases FLIR A6901scSLS (°C)
1	Ballistic barrel 7.62 × 51 mm	7.62 × 51 mm SWISS PAP	164.2	<150
2			270.4	-
3		7.62 × 51T	337.1	386.5
4			850.5	229.9
5		7.62 × 51 mm M80	138.6	127.2
6			337.9	149.5
7	Ballistic barrel 7.62 × 54R	7.62 × 54R B-32	146.2	262.1
8			700.8	<150
9			84.4	263.3
10		7.62 × 54R LPS	839.5	721.1
11			315.5	165.4
12			725.8	285.6
13	Ballistic barrel 7.62 × 51 mm	.308 Win. Norma Ecostrike	272.9	95.7
14			274.6	<150
15	Ballistic barrel 5.56 × 45 mm	5.56 × 45 mm SS109	315.3	<150
16			951.4	165.4
17		5.56 × 45 mm M193	150.9	<150
18			150.8	<150

- Not applicable.

**Table 2 materials-15-04693-t002:** Thermal energies during projectiles ricochetting on the Armox 600 armour plate.

No.	Type of Projectile;Kinetic Energy (J)	ProjectileCross-Section	Maximum Temperature (°C)	Thermal Energy (J)
1	7.62 × 51 mm SWISS PAP; 3963	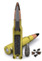	165–270	1600–2000
2	7.62 × 51T; 3097	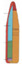	340–850	2000–3700
3	7.62 × 51 mm M80; 3455	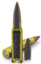	140–340	740–1120
4	7.62 × 54R B-32; 3412	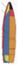	140–840	3400–9500
5	7.62 × 54R LPS; 3287	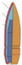	315–840	1100–2100
6	.308 Win. Norma Ecostrike; 3589	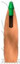	>273	>1025

**Table 3 materials-15-04693-t003:** Thermal energies during the ricochetting of projectiles on the surface of non-metallic components.

No.	Type of Projectile	Ricochet Material (Thickness (mm))	Maximum Temperature (°C)	Thermal Energy (J)
1	7.62 × 54R LPS	concrete sidewalk slab (70)	165.3	800
2	7.62 × 54R B-32	concrete sidewalk slab (70)	216.9	980
3	7.62 × 54R B-32	gray granite slab (60)	216.3	980
4	7.62 × 54R B-32	gray granite slab (60)	219.0	990
5	7.62 × 54R B-32	concrete block (120)	272.1	1100
6	7.62 × 54R B-32	concrete block (120)	323.8	1200
7	7.62 × 54R B-32	stoneware gres on concrete slab (gres 8) (concrete 40)	327.5	1200
8	7.62 × 54R B-32	stoneware gres on concrete slab (gres 8) (concrete 40)	323.4	1200
9	7.62 × 54R B-32	black granite slab (57)	325.1	1200
10	7.62 × 54R LPS	black granite slab (57)	325.0	1100

**Table 4 materials-15-04693-t004:** Flash points and minimum ignition energies for the selected flammable gases and vapours of liquids.

No.	Name of Substance	Chemical Formula	Molecular Weight (g/mol)	Flash Point (°C)	Minimum Ignition Energy (mJ)
1	acetone	CH_3_COCH_3_	58.1	540	0.25
2	acetylene	C_2_H_2_	26.0	305	0.011
3	ethyl alcohol	C_2_H_5_OH	46.1	425	0.4
4	methyl alcohol	CH_3_CH(OH)CH_3_	60.1	400	0.65
5	isopropyl alcohol	CH_3_CH(OH)CH_3_	60.1	400	0.65
6	automotive gasoline	-	95.3–98.2	300	0.15
7	n-butane	C_4_H_10_	58.1	430	0.25
8	methane	CH_4_	16.04	650	0.28
9	kerosene	-	-	>250	0.65
10	gas oil	-	-	250	0.48
11	propane	C_3_H_8_	44.1	500	0.22
12	trichloroethylene	ClCH=CC_l2_	131.4	410	300
13	hydrogen	H_2_	2.016	580	0.018

- Not applicable. Source: own elaboration.

## Data Availability

Not applicable.

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
