# Peer review of "Thermal Energy Analysis of Projectiles during Ricochetting Using a Thermal Camera"

_materials, 2022, doi:10.3390/ma15134693_

Round 1

Reviewer 1 Report

Interesting paper as a methodology to assess the ballistic impact. But paper should be consider only a proposal of a methodology because, statistically, there are too few fires to support the conclusions and some results are too spread to be reliable. Also, I suppose that differences could be produced by the impact velocity which was not measured for each fire, the authors considering the datasheet information for each projectile.

I suggested several comments that authors could do in order to make the paper clearer in the attached document. 

Author Response

The paper was corrected, the corrections were appropriate marked.

The authors extend their appreciation the reviewer for his valuable comments.

Reviewer 2 Report

The article presents experimental tests of small caliber projectiles that strike different types (materials) of targets. The aim of the study is to calculate the thermal (ignition) energy produced from the expulsion of the projectile and the target using a high-speed camera and thermal cameras that are associated with image processing software. The article needs extensive and serious revisions to be considered for publication in Materials. Following are the recommendations to improve the quality of the article.  

1-     More details about the test methodology are required to be summarized in the abstract.

2-     Nothing was presented in the abstract about the obtained results and concluded remarks. The abstract should be a stand-alone part of the article that briefly presents the aim of the study, the methodology and the most important results. It is strongly recommended to end the abstract with at least two sentences that summarize the most important concluded remarks. Adding the significant obtained numeral comparisons at the end of the abstract is also recommended.

3-     The introduction section is too limited and does not provide the readers with the background knowledge of the investigated subject. This section must be improved significantly by conducting the followings; (a) instead of addressing mass citations for limited general statements as in lines 37 and 38 (i.e. [1-7] and [8-23]), more detailed and more beneficial background knowledge from these references should be added to this paragraph with more proper citation. (b) rich but brief literature review of the leading and most recent close research works should be added. The used methodologies and the most important conclusions of the reviewed researchers should be presented. This literature review should explore and present the state-of-the-art available knowledge in the investigated field of study. (c) based on the reviewed literature review, the authors should present in the last paragraph of the introduction section the gaps of knowledge in this field. (d) Most importantly, the last paragraph should explicitly highlight the novel points of the presented work over the available literature and what gaps of knowledge are planned to be filled. Unfortunately, none of the above addressed points is presented in the current version of the article. Therefore, it is highly recommended to follow the above mentioned recommendations to improve the final quality of the article.

4-     The scale bars shown at the right of the pictures presented in Figures 4 and 5 are not readable and the unit is not shown. Similarly, what are the units of the scale bars shown in Figures 8 and 9?

5-     In sections 2.1 and 2.2, the apparatus and materials used to conduct the two tests were listed without any details of specifications (or very limited) and without addressing the function of each element of the test apparatus and why it was chosen. It is recommended to address more details about each of the test elements, test apparatus and test materials used. For instance, concrete elements can be composed of different types of concrete that have a wide range of projectile impact resistance and high-temperature performance. Similarly, reinforced concrete elements behave completely differently from plain ones under all types of loads including projectiles. More details about the concrete type, its mix materials, strength, and reinforcement should be provided, and so as for the other adopted test targets.

6-      It is not clear if the shape of thermal energy propagation shown in Figure 11(a) can be classified as a semi-cone. For this reviewer, the irregular volume shown in Figure 11(a) cannot be considered as a semi-cone. Similarly, it is somewhat difficult to say that the shape shown in Figure 11(b) can be approximated to a semi-sphere. To better visualize the idea of the authors and their visual inspection remarks, border shapes of a cone and a sphere should be sketched over the recorded thermal energy propagation volumes to visualize the similarity with these standard shapes.

7-     Figures 12 and 13 are not suitable for final production in their current presentation. Yes, these are the outputs of the used software, but the data can be taken and better visualized using any graphing software. The number of ticks in the vertical and horizontal axes should be reduced and the font size of tick labels and axes titles should be increased. The unit of the temperature should also be added in the vertical axis.

8-     The discussion of Figure 13 is too limited, while there is no discussion at all for Figure 12. The authors should discuss the two obtained scenarios in terms of the causes of peak temperatures and the time of the ascending and descending parts before and after the peak temperatures.

9-     As for the previous notice, the discussion of the results presented in Table 1 is too limited. Similarly, the much important results of thermal energy presented in Tables 2 and 3 lag for a serious discussion.  In general, the discussion of all sections and all obtained results is inadequate. This section should seriously be improved with deep discussions that consider the effect of all tests and materials effective factors. The authors should also show some comparisons (agreement or disagreement) with related literature results, especially if different techniques were used to measure the thermal energy.

10- Line 201 “The thermal energy calculated in Chapter 3” what chapter?

Author Response

Ref. 1 and 2. Abstract was expanded to include new information on the munitions used and the main conclusions.

Ref. 3. The introduction was expanded by adding a brief summary of the paper [8] that described a similar type of study. More references describing this type of study was not found. It was pointed out that there are too small papers in this matters.

Ref. 4: In Figures 4, 5, 8 and 9 the font was increased and units were added.

Ref. 5: More detailed information on the apparatus used and samples were added.

Ref. 6: A sketch of the shape of a semi-cone and a semi-sphere were added.

Ref 7. In Figures 12 and 13 the fonts were increased.

Ref 8. Add new comments on Figures 12 and 13.

Ref.9. Additional comments were added to describe data in Table 1.

Ref.10. “Section” instead of “Chapter”

The authors extend their appreciation the reviewer for his valuable comments.

Round 2

Reviewer 2 Report

The authors did not strictly follow the recommendations of the previous review round. However, the article can be accepted in its current state.